# A Cluster Randomised Control Trial of an SMS-Based Intervention to Promote Antenatal Health amongst Pregnant Women in a Remote, Highland Region of Vietnam

**DOI:** 10.3390/healthcare11172407

**Published:** 2023-08-28

**Authors:** Binh Thi Thanh Dao, Huong Thi Trinh, Phuong Hoai Lai, Tahli Elsner, Sumit Kane, Liem Nguyen

**Affiliations:** 1Faculty of Management and Tourism, Hanoi University, Hanoi 100000, Vietnam; lhphuong@hanu.edu.vn; 2Faculty of Mathematical Economics, Thuongmai University, Hanoi 100000, Vietnam; 3Nossal Institute for Global Health, Melbourne School of Population and Global Health, The University of Melbourne, Melbourne, VIC 3010, Australia; tahli8597@gmail.com (T.E.); sumit.kane@unimelb.edu.au (S.K.); 4Institute of Population, Health and Development, Hanoi 100000, Vietnam; ntl1234@gmail.com

**Keywords:** mHealth, antenatal care, maternal health, health-seeking behaviour, Vietnam

## Abstract

Although Vietnam has achieved significant improvements in maternal, newborn, and children’s health, outcomes for ethnic minorities living in remote mountainous areas continue to lag. Interventions that leverage the extensive mobile networks in the country have been proposed as a way to overcome some of these challenges. A cluster randomised controlled trial (cRCT) was conducted to assess the effectiveness of an intervention comprising tailored SMS messages for promoting antenatal care knowledge and behaviours amongst ethnic minority (EM) pregnant women. The cRCT was implemented across eight intervention communes (640 women) and four control communes (315 women) in Northern Vietnam. Maternal health-related knowledge and behaviour outcomes and self-rated health status were assessed through questionnaires administered pre- and post-intervention. Difference-in-difference and logistic regression analysis found that the intervention group showed significant improvements in awareness about the danger signs of pregnancy and the importance of nutritional supplements. Significant improvements were seen in antenatal care-seeking behaviours and the intake of nutritional supplements. Mobile messaging-based behaviour change interventions can significantly improve maternal health-related knowledge and care-seeking amongst women residing in marginalised, hard-to-reach populations.

## 1. Introduction

Vietnam has achieved significant improvements in maternal, newborn, and children’s health (MNCH) outcomes. Between 1990 and 2020, infant mortality fell from 37 to 17 deaths per 1000 live births, and under-five mortality went from 52 to 21 [1] and maternal mortality from 139 to 124 [2]. However, significant inequities in MNCH outcomes remain. In Vietnam, the Kinh ethnic group makes up the majority of the population, whereas a total of 53 ethnic minority (EM) communities make up an estimated 19% of the population [3]. EM groups have poorer population health than the Kinh majority [3,4]. For example, the maternal mortality ratio amongst ethnic minority groups is four times higher than amongst the majority Kinh ethnic group [5]. Research suggests that this is primarily because EM communities disproportionately live in remote, rural areas where access to health services is constrained [4]. Figure A1 shows how health service coverage and utilisation indicators are on average 30% lower amongst ethnic minorities of Vietnam. The Government of Vietnam and non-government organisations have been trying various approaches to help improve MNCH service utilisation and outcomes for ethnic minority communities but have had limited success. For instance, evaluations of some MNCH-focused interventions in the remote, mountainous Dinh Hoa district found that ethnic minority communities experienced the information education communication (IEC) approach as being passive, impersonal, and unidirectional and found the interventions to be lacking in specificity, detail, and practical applicability for their local context [6,7]. Although there are many strategies to improve the use of maternal health services in low- and middle-income countries [8], evidence [7,9] points to the potential of mobile telephony-based (mHealth) interventions. There is robust evidence around the effectiveness of mHealth interventions to help promote healthy behaviours, to remind populations about preventive and protective measures, and to facilitate access to and the use of health services [10] globally and in Vietnam. However, the evidence on the effectiveness of mHealth interventions in marginalised communities living in hard-to-reach areas is limited.

Given the above, we set out to develop, implement, and test an mHealth intervention (a mobile health intervention for mothers and children, which is abbreviated as mMoM) to address these constraints and gain evidence. In view of the ubiquity of mobile phone use across the whole of Vietnam, including amongst ethnic minority communities, mobile phone-based approaches have been proposed as a potentially effective way to improve access to and for health services to better connect and engage with the population, particularly with hard-to-reach communities [11,12,13]. We developed, implemented, and tested the mMoM intervention, a short message service (SMS)-based approach in Dinh Hoa district—a district where ethnic minorities account for 73.6% of the population [4,14]. To our knowledge, mMom is a first-of-its-kind study examining the effectiveness of a mobile phone-based intervention for promoting healthy behaviours amongst pregnant women and new mothers from a hard-to-reach, marginalised community in a low- and middle-income country context.

The mMoM intervention took a behaviour change communication approach and involved the delivery and exchange of SMS messages with pregnant women, providing actionable advice on how to remain healthy, optimise foetal development during pregnancy, and implement appropriate care (antenatal, delivery, and postnatal care) [15]. The details of and the processes involved in the development of the mMoM intervention are presented in a previously published protocol article [4]; the intervention components were:Prenatal SMS program consisting of 75 SMS messages delivered between weeks 5 and 42 of pregnancy, two to three times per week, covering topics according to the stage of pregnancy.mMOM database integrated with existing health information and registration systems. Each woman has a unique identification code (UIC) that allowed messages to be tailored to her stage of pregnancy and the state of interaction with the health system.Messages included one-way informational/educational messages, reminder messages, and interactive messages.In the event of a woman not responding to or not following up on the message (one reminder), an SMS was sent to a commune health worker (CHW) to initiate in-person follow up.

The mMoM intervention implementation was set up as a cluster randomised controlled trial (cRCT) to assess the impact of the intervention on the MNCH knowledge, behaviour, and self-reported health status of pregnant women living in Dinh Hoa. It was expected that SMS-based communication would help overcome the major communication barriers that EM women face in accessing MNCH information and in using MNCH services. Specifically, we hypothesised that mMoM would improve MNCH awareness and knowledge, particularly regarding the danger signs of pregnancy. We hypothesised that mMoM would positively impact MNCH behaviours such as the intake of supplements, attendance at monthly blood pressure and weight check-ups, and undergoing the necessary blood and urine tests. We also hypothesised that mMoM would lead to an overall improvement in self-rated health status amongst participating women. This paper reports on the effects of the prenatal or pregnancy part of the mMoM intervention.

## 2. Materials and Methods

Ethical approval for this study was granted by the independent institutional review board at the Institute for Population Health and Development, Hanoi (PHAD). The board is registered with and uses the ethics review protocols of the Office for Human Research Protections of the United States Department of Health and Human Services. To ensure privacy and to maintain confidentiality, all personal data were anonymised and delinked from the participants prior to the analysis. The protocol of this program is also available [4].

### 2.1. Study Design

A cRCT (cluster randomised controlled trial) was conducted to study the effects of the mMoM intervention on MNCH-related knowledge, behaviours, and health status of participating pregnant women. It was assumed that women living in the same, often small, and tight-knit community would be well acquainted with each other, would communicate regularly, and could pass on their MNCH knowledge and behaviour; hence, a cluster design was adopted over a simple random design. Dinh Hoa consists of 24 communes; in this trial, a cluster comprised pregnant women from a single commune. mMoM and the cRCT were implemented across 12 communes—8 intervention communes and 4 control communes. As our hypothesis was that the mMoM intervention would positively influence MNCH outcomes, we selected more communes to receive the intervention. The cRCT was conducted from 1 July 2014 until 30 September 2016 (Figure 1).

All 24 communes in the Dinh Hoa district were listed and stratified into 3 groups: central (i.e., around district centre), midway, and distant or hard to reach. Using stratified randomisation, 12 out of the 24 communes were selected for the study; intervention and control communes were selected from each stratum using probability proportional to size sampling (PPS). From the 8 central communes, 2 communes were included in the intervention group and 1 in the control group; among the 11 middle communes, 3 were assigned the intervention and 1 the control; and among the 5 hard-to-reach communes, 3 were selected as intervention communes and the remaining 2 were selected as control communes. The pregnant women in the intervention communes received the mMoM intervention in addition to the existing, routine MNCH services, whereas women in control communes only received existing, routine MNCH services and did not receive the intervention.

### 2.2. Sample

The eligibility criteria for study participants were (a) de-facto resident of a study commune and (b) currently pregnant. Women were recruited by commune health workers (CHWs) who, given their local knowledge and relationships, were able to identify all the pregnant women in their communes and invite them to participate in the study. Study participants completed two interviews: once at the time of recruitment when they were known to be pregnant (“pre-pregnancy”), and once after delivery, after miscarriage, or upon exiting the study for any other reason (“post-pregnancy”). In total, 860 women (547 intervention and 313 control) were initially recruited to participate in the study and completed the “pre-pregnancy” survey upon recruitment. Of these, 525 and 307 women were from the intervention and control communes, respectively. Less than 5% of the initial participant pool had incomplete data and were excluded from the analysis.

### 2.3. Measures

Three sets of primary outcome measures were assessed. First was changes in MNCH knowledge. The three MNCH knowledge outcomes assessed were women’s awareness of the danger signs during pregnancy, understanding of the importance of antenatal supplements, and knowledge of their blood type. Participating women were asked “*Can you name the danger signs during pregnancy?*” Unprompted responses were recorded under eight major danger signs during pregnancy (bleeding, swelling, high fever, convulsion, leaking amniotic fluid, difficulty breathing, abdominal pain, painful urination) or under “*Others*” or “*Do not know*”. Participants were also asked to name supplements that would be useful during pregnancy. Unprompted responses were recorded for four key antenatal supplements (folic acid, iron, calcium, and docosahexaenoic acid, i.e., DHA) or under “*Using supplement but cannot name it*” or “*Do not know*”. Women who took a blood test were asked whether they knew their blood type. Second was MNCH behaviour outcomes. The five MNCH behaviour outcomes measured included supplement intake during pregnancy, monthly blood pressure check, monthly weight check attendance, blood test attendance, and urine test attendance. These outcomes were measured by asking women whether they took certain supplements or whether they attended these check-up or test appointments in pre- and post-pregnancy surveys. Third was self-reported health status. This was measured by asking women to rate their health status at the time of interview as either “*very good*”, “*good*”, “*normal*”, “*poor*”, or “*very poor*”.

### 2.4. Analysis

Difference-in-difference estimations (DiD) were used to analyse the net effect of the mMoM intervention on the MNCH outcomes. The difference between MNCH outcomes of participants recorded pre- and post-pregnancy was calculated for both the control and the intervention cohort. These differences were then compared to generate a difference of the differences. DiD was chosen for analysis to account for the bias of a simple post-pregnancy comparison between the intervention and control communes, i.e., bias due to fixed and pre-existing differences between the control and intervention cohorts; it also removes the bias potentially generated by trends impacting the study population over time [16]. DiD was implemented as an interaction term between time and intervention group dummy variables in a regression model:(1)$$Y=\beta_{0}+\beta_{1}D_{mMOM}+\beta_{2}D_{Post}+\beta_{3}\left({D_{mMOM}\times D}_{Post}\right)+\beta_{4}\left(Covariates\right)+\varepsilon$$
where *Y* is an MNCH outcome, $D_{mMOM}$ captures possible differences between the intervention groups ($D_{mMOM}=1$) and control groups ($D_{mMOM}$ = 0) prior to the mMoM intervention, and $D_{Post}$ captures aggregate factors that would cause changes in *Y* even in the absence of the mMoM intervention ($D_{Post}=0\;$means before intervention, $D_{Post}=1\;$means after intervention). Model (1) is a logistic regression.

Following [17], the estimated DiD is the $\hat{\beta_{3}}$ coefficient in Equation (1), i.e.,
(2)$$\hat{\beta_{3}}=\left({\bar{Y}}_{mMOM,\;Post}-{\bar{Y}}_{mMOM,\;Pre}\right)-\left({\bar{Y}}_{Control,\;Post}-{\bar{Y}}_{Control,Pre}\right)$$

Data were analysed using R version 4.1.3, and DiD was estimated using the glm function (binomial family) in the *stats* package. We used logistic regression whilst controlling for age, ethnicity, level of education, working status, household income, household size, health status, and trimester of pregnancy at the time of entering the study.

The results from the descriptive analysis (*survey* package) and the effect on the focus variables (i.e., health knowledge, behaviour, and health status) are presented in the next section, and the results of selective regression models predicting supplement awareness with all control variables are presented in Appendix A (Table A1, Table A2, Table A3, Table A4 and Table A5).

False discovery rate (FDR) correction for multiple comparisons was applied [18,19] (using the *p.adjust()* function in R version 4.1.3). A two-tailed FDR-corrected *p* < 0.05 was considered statistically significant.

## 3. Results

### 3.1. Descriptive Statistics

Pregnant women in the intervention and control groups had very similar demographic characteristics. A total of 76.4% of the study population identified as an ethnic minority (EM); this is reflective of the broader population in Dinh Hoa district, which is 73.6% EM according to the 2019 Population and Housing Census [14]. The mean age of the study participants was 27.7 and 27.9 for the intervention group and control group, respectively. A total of 65.8% had completed secondary school education, and 95% had health insurance (typically under the government-sponsored scheme for economically disadvantaged and EM people). Almost all study participants rated their health as normal (i.e., average) or higher (Table 1).

### 3.2. Knowledge

#### 3.2.1. mMoM’s Effects on Awareness of Danger Signs during Pregnancy

The mMoM intervention significantly improved awareness of five out of the eight danger signs (swelling, high fever, convulsion, leaking amniotic fluid, and abdominal pain) of pregnancy among participating women. However, mMoM did not have a significant impact on increasing recognition of painful urination, difficulty breathing, and bleeding as danger signs among the participants. It should be noted, however, that awareness of bleeding was already high at the pre-pregnancy stage among participants in both the control and intervention groups, and the increase in awareness of painful urination and difficult breathing as danger signs was greater (although not statistically significant) in the intervention group than the control group (Table 2).

#### 3.2.2. mMoM’s Effects on Awareness about Nutritional Supplements

The mMoM intervention significantly improved awareness of all four pregnancy-related nutritional supplements assessed in this study (iron, folic acid, DHA, and calcium). Iron was the supplement that women had the most pre-existing knowledge of, with exceptionally high levels of awareness among all participants at the pre-pregnancy stage. Despite the mMoM intervention significantly improving women’s awareness of DHA and folic acid as supplements that should be taken during pregnancy, the proportion of women aware of these two supplements at the post-pregnancy stage remained much lower than awareness of iron and calcium (Table 3). In addition, it was found that ethnicity was significantly associated with supplement knowledge (Table A1, Table A2, Table A3, Table A4 and Table A5) For example, comparing to the Kinh ethnic group, Tay women were significantly less likely to be able to name folic acid (Table A1), but they were more likely to be able to name calcium (Table A3) as a supplement and also less likely to be unable to name a supplement that they were using (Table A5). Nung women were also less likely to be able to name folic acid as a supplement (Table A1). Mothers who had higher levels of education had significantly better awareness of all four supplements than those with a junior secondary or lower level of education (*p* ≤ 0.001) (Table A1, Table A2, Table A3 to Table A4). For example, the likelihood of folic acid awareness amongst pregnant women with a senior high school education was more than two times as much as that of those with only a junior secondary or lower level of education (adj. OR: 2.20 95% CI [1.74–2.78], *p* ≤ 0.01) (Table A1).

### 3.3. Behaviour

#### 3.3.1. mMoM’s Effect on Prenatal Care Behaviours

The mMoM intervention significantly improved four out of the five prenatal care behaviours that were targeted (monthly weight checking, taking a blood test, knowing blood type, and taking a urine test). mMoM did not significantly affect the monthly blood pressure-checking behaviour; this may be because most women could not afford a blood pressure monitor at home and/or travelling to the CHC every month to have their blood pressure checked. Although the majority of women took a blood test, only a small proportion of them knew their blood type. This result may imply that the SMS messages provided through the mMoM intervention on blood type were not enough to improve awareness of blood type, that the notion of a blood group is not something that is well understood in the community, or that the nomenclature used to describe blood groups and types in the SMS messages may not be easily understandable in the local context (Table 4).

#### 3.3.2. mMoM’s Effect on Supplement Intake

The effects of mMoM on supplement intake strongly correlated with the effects on improving supplement awareness; the intervention significantly increased the proportion of women taking folic acid, DHA, and especially calcium. mMoM did not show a significant effect on iron intake; however, this is reasonable given the already high proportion of women taking iron regardless of mMoM. Despite mMoM significantly improving DHA and folic acid intake, the proportion of women taking these supplements in both the control and intervention cohorts remained low in comparison to calcium and iron intake (Table 5).

#### 3.3.3. mMoM’s Effects on the Self-Reported Health Status of Women

Being a recipient of the mMoM intervention was associated with significant improvement in the self-rated health status of pregnant women. A clear difference between the control and the intervention group was found: Women in the control group were less likely than women in the intervention group to rate their health as good or very good (Table 6).

## 4. Discussion

Although the evidence of the effectiveness of mobile phone-based interventions in promoting healthy behaviours and in facilitating access to and the use of maternal health services in remote areas of LMICs is robust [7], to our knowledge, this is the first study to show these effects in marginalised communities. Consistent with existing evidence [7], the effects of mMoM were significant where the MNCH outcomes were initially poorer, and vice versa. For instance, mMoM did not have a significant effect on outcomes such as iron intake and awareness of bleeding as a danger sign during pregnancy; this may have been because baseline levels of iron intake and awareness of bleeding among study participants were high prior to the study commencing. That said, mMoM did not have a significant effect on the awareness of shortness of breath and painful urination as danger signs of pregnancy or on improving the monthly blood pressure-checking behaviour. The lack of significant effect on these outcomes, which cannot similarly be explained by high pre-existing levels of awareness or compliance, suggests that there may be aspects unique to the study population that need to be understood and addressed in a contextually appropriate manner. The importance of contextualising mHealth interventions was highlighted by Paduano et al. (2022) in Tanzania and by Benski et al. (2020) in Madagascar. More generally, though, the limits of an SMS-based approach in improving MNCH indicators in this setting are clear from the lack of improvements in some outcomes, such as awareness of swelling and convulsions as danger signs, DHA as an important supplement during pregnancy, and knowledge of blood type [9,21]. Evidence suggests that multipronged approaches that involve messages, reminders, and coaching sessions that enabling health workers to support women, together with in-person education [10], have the potential to have a greater effect on MNCH-related behaviour change.

The study revealed a strong association between awareness of the importance of supplements and their eventual intake. Women who were aware of a supplement were more likely to take it, which is an effect that had even greater influence on intake than the ability to pay (assuming a woman’s working status is a proxy of ability to pay). This direct link between knowledge and behaviour emphasises the importance of the health education component of behaviour change communication interventions. Interestingly, although most women in both the control and intervention cohorts had taken a blood test by the end of their pregnancy, a much smaller proportion of them knew their blood type. This may imply that this relationship between knowledge and behaviour may not be bidirectional and that eliciting the desired behaviour may not necessarily translate to increased knowledge and awareness. As indicated earlier, it is also possible that the notion of a blood group and its importance are not something that was well understood in the community, or perhaps the nomenclature used to describe blood groups and types may not have been easily understandable.

Despite mMoM’s positive effects on MNCH knowledge, some behaviours, and the self-reported health status of pregnant women in the study setting, the mHealth intervention’s performance was inevitably constrained by issues such as unreliable or poor mobile network coverage, electricity supply, mobile phone maintenance, user proficiency, and the literacy of users [22,23]. These limitations highlight that regardless of an mHealth approach’s ability to address accessibility issues faced by hard-to-reach populations and enhance utilisation of MNCH services, it cannot yet replace high-quality, responsive, and accessible in-person MNCH health promotion and prevention services.

The study design also had some limitations. The study included a limited set of outcome measures, with only areas of knowledge regarding danger signs, supplements, and blood type being assessed and behaviours such as supplement taking and prenatal check attendance and testing being assessed. Women’s awareness and behaviours related to other key MNCH areas such as maternal nutrition, postnatal health, birth-spacing and contraception, breastfeeding and lactation, and infant immunisation were not assessed. Further, the study only measured self-reported health status and could not assess the effect on more objective MNCH health measurements such as maternal iron-deficiency anaemia, incidence of perinatal complications, preterm birth, low birth weight, length of time breastfeeding, or maternal or infant mortality and morbidity. A wide range of health and behavioural outcomes can potentially be influenced by intervening through mobile phone-based messaging; this requires further investigation.

Inequities in MNCH and broader development outcomes between ethnic minorities and the Kinh majority in Vietnam are well documented. However, the qualitative fieldwork conducted in the study area (Dinh Hoa district) prior to the trial’s commencement found no significant differences between socioeconomic or health status of ethnic minority and Kinh women. Similar findings emerged during the baseline assessments, too; the exception was the San Chay ethnic minority. In a population where baseline differences between ethnic minority and Kinh women are more pronounced, the impact of mMoM-like interventions would likely be more pronounced. Similarly, a larger study population with a range of minority communities would allow one to obtain a better picture of the heterogeneities across ethnic minorities [3,5]. As the study was limited to a single district in a single province in Vietnam, the generalisability of its findings may be limited to the Dinh Hoa district or other districts with similar geographic, demographic, and economic circumstances. It should also be noted that fieldwork conducted in 2013, i.e., prior to the study’s commencement, found that antenatal care and facility-based delivery in the Dinh Hoa district was already high, with the majority of women attending three or more antenatal visits, 100% of deliveries (in the first half of 2013) occurring at a facility, and only a single home delivery occurring in the year prior. The positive effects of an intervention like mMoM may not be replicated in differing contexts where women have a lower pre-existing level of interaction with the health system [4].

## 5. Conclusions

The previous results of the impact of mHealth interventions on MNCH in low- and middle-income countries (LMICs) reveal that mHealth interventions are effective at improving various MNCH outcomes such as increased utilisation of antenatal and postnatal care (PNC) services, use of skilled birth attendance, immunisation, iron supplementation, and exclusive breastfeeding [7,10,24,25,26,27,28,29]. Mbuthia et al. (2019) [30], in their systematic review of the impact of mHealth-based interventions on PNC in rural settings, found that SMS-based messaging can positively influence attitudes, norms, skills, and self-efficacy among mothers and address environmental barriers that hinder the uptake of PNC in rural areas [30]. Despite mHealth interventions showing promise in improving MNCH outcomes, existing evidence is still of modest certainty due to most studies being of low or moderate quality; the need to conduct more high-quality and methodologically robust studies that report across a wider variety of settings and MNCH health outcomes to strengthen evidence in this area has been noted [7,30].

To our knowledge, mMoM is the first randomised controlled trial to evaluate the effectiveness of an SMS-based health promotion intervention that specifically targets improving MNCH outcomes among marginalised ethnic minority women in Vietnam. We found that a multipronged behaviour change-focused intervention approach consisting of a package of SMS messages tailored to individual needs has the potential to improve MNCH-related knowledge, behaviours, and self-perceived health status among ethnic minority pregnant women in Vietnam.

In LMICs such as Vietnam, mobile phone-based health promotion interventions can be used to enhance the reach and responsiveness of existing services, especially among hard-to-reach marginalised populations that often experience greater barriers to accessing care [3,13,31]. Combining higher-quality in-person care with mobile phone-based interventions like mMoM while integrating them into existing health systems has the potential to yield good outcomes for hard-to-reach populations. Future research should explore the effects of interventions such as mMoM on a larger scale, across a broader range of MNCH and other health outcomes, and across a wider range of ethnic minority groups in Vietnam and beyond.

## Figures and Tables

**Figure 1 healthcare-11-02407-f001:**
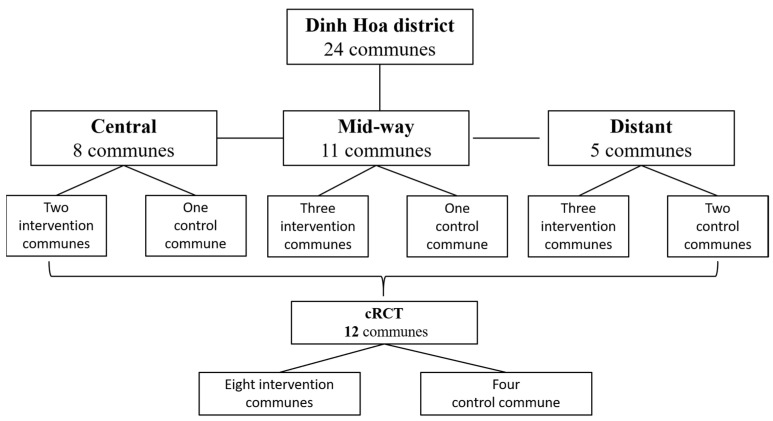
Study flow diagram.

**Table 1 healthcare-11-02407-t001:** Demographic and socioeconomic characteristics of pregnant women.

Variables		Intervention Groups		Control Groups		*p*-Value
Namevar		Mean	95% CI	Mean	95% CI	
Age		27.7	[27.3–28.1]	27.9	[27.3–28.6]	0.57
Ethnicity (%)	Kinh	23.6	[20.1–27.2]	26.8	[21.3–32.6]	0.28
	Tay	52.0	[47.8–56.3]	56.7	[50.4–62.9]	0.37
	Nung	4.9	[3.1–7.0]	6.0	[3.3–9.3]	0.88
	San Chay	10.7	[8.3–13.5]	4.8	[3.0–7.1]	<0.01
	Other	8.8	[6.6–11.3]	5.8	[3.2–9.0]	0.25
Education level (%) (completed)	Junior secondary or lower	34.2	[30.3–38.3]	32.2	[26.8–37.9]	0.88
	Senior high school	44.8	[40. 6–49.1]	52.2	[46.0–58.4]	0.25
	University and college or vocational school	21.0	[17.6–24.7]	15.5	[11.2–20.4]	0.08
Working status (%)	Yes	66.1	[62.4–69.8]	49.8	[44.5–55.1]	<0.01
	No	33.9	[30.2–37.6]	50.2	[44.9–55.5]	<0.01
Per capita income (VND)		1,167,373	[1,048,197– 1,286,548]	1,317,404	[1,109,701– 1,525,108]	0.28
Household income (VND)		3,580,984	[3,324,986– 3,836,982]	3,158,098	[2,870,177– 3,446,019]	0.05
Household size		4.0	[3.9–4.1]	4.0	[3.9–4.0]	0.28
Current height		153.8	[153.4–154.2]	154.0	[153.4–154.6]	0.99
Current weight		50.6	[50.0–51.2]	50.8	[50.0–51.6]	0.90
Weight before pregnant		46.3	[45.9–46.8]	45.8	[45.1–46.5]	0.28
Health insurance (%)	Yes	95.0	[93.0–96.7]	97.9	[95.9–99.2]	0.19
	No	5.0	[3.3–7.0]	2.2	[0.8–4.1]	0.19
Health Status (%)	Very good	1.0	[0.3–2.0]	1.7	[0.5–3.4]	0.88
	Good	56.2	[52.1–60.3]	42.3	[36.4–48.2]	<0.01
	Average	41.2	[37.1–45.3]	52.3	[46.3–58.3]	0.02
	Poor	1.6	[0.8–2.7]	3.8	[1.6–6.8]	0.25

Source: authors’ calculations. Note: Means are reported for continuous variables and percentages are reported for discrete variables. The *p*-value shows the significant mean difference between the control and the intervention groups; a conventional cut-off value at 5% or *p* ≤ 0.05 is used to identify significant difference. All the numbers have been adjusted to the sample design using the package survey in R software version 4.1.3 [20]. The *p*-values are corrected for multiple comparisons using FDR correction.

**Table 2 healthcare-11-02407-t002:** Proportion of pregnant women who could identify a specific danger sign when asked “*Can you name the danger signs during pregnancy*?”.

Danger Sign	Control			Intervention			Raw DiD	Adj. DiD	
	Before	After	Diff.	Before	After	Diff.		Diff.	*p*-Value
	(1)	(2)	(3) = (2) − (1)	(4)	(5)	(6) = (5) − (4)	(7) = (6) − (3)	(8)	
Bleeding	78.3	89.6	11.3	74.7	89.9	15.2	3.9	1.42	0.28
Swelling	23.0	30.3	7.3	25.2	42.7	17.4	10.1	2.57	<0.01
High fever	39.6	34.9	−4.7	45.2	59.9	14.7	19.4	3.12	<0.01
Convulsion	23.4	33.2	9.8	28.0	40.5	12.5	2.7	1.58	0.05
Leaking amniotic fluid	29.1	38.5	9.4	30.4	53.9	23.5	14.1	1.95	<0.01
Difficult breathing	9.7	17.6	7.9	22.0	30.9	8.9	1.1	1.38	0.28
Abdominal pain	65.7	61.4	−4.4	67.3	80.8	13.5	17.9	2.91	<0.01
Painful urination	1.9	5.0	3.1	8.7	20.5	11.7	8.6	1.82	0.3
Do not know	8.2	2.3	−5.9	10.8	0	−10.8	−4.8	0.05	0.02

Source: authors’ calculations. Note: Adj. DiD = adjusted DiD; column 8 presents $\hat{\beta_{3}}$ coefficients from regression model (1) predicting DiD of awareness of a danger sign during pregnancy after controlling for other covariates, including age, ethnicity, education, working status, household income, household size, number of children, and health status. *p*-values are corrected for multiple comparisons using FDR correction.

**Table 3 healthcare-11-02407-t003:** Proportion of women who could name a specific supplement when asked to “*Name all the supplements you think that women should take during pregnancy*”.

Supplement	Control			Intervention			Raw DiD	Adj. DiD	
	Before	After	Diff.	Before	After	Diff.		Diff.	*p*-Value
	(1)	(2)	(3) = (2) − (1)	(4)	(5)	(6) = (5) − (4)	(7) = (6) − (3)	(8)	
Folic acid	44.9	59.0	14.1	38.7	64.8	26.2	12.0	2.53	<0.01
Iron	95.5	96.7	1.2	92.1	98.9	6.8	5.6	4.8	0.01
Calcium	85.3	75.6	−9.7	68.1	83.6	15.5	25.2	6.45	<0.01
DHA	39.4	22.3	−17.1	36.3	46.7	10.4	27.5	5.26	<0.01
Using supplement but cannot name it	3.5	2.5	−1.0	1.3	2.3	1.0	2.0	2.58	0.15
Do not know	2.7	1.7	−1.0	6.3	0.1	−6.1	−5.1	0.07	0.03

Source: authors’ calculations. Note: Adj. DiD = adjusted DiD; column 8 presents $\hat{\beta_{3}}$ coefficients from regression model (1) predicting DiD of supplement awareness after controlling for other covariates, including age, ethnicity, education, working status, household income, household size, number of children, and health status. The *p*-values are corrected for multiple comparisons using FDR correction.

**Table 4 healthcare-11-02407-t004:** Proportion of women engaging in certain prenatal care behaviours.

Prenatal Care Behaviours	Control			Intervention			Raw DiD	Adj. DiD	
	Before	After	Diff.	Before	After	Diff.		Diff.	*p*-Value
	(1)	(2)	(3) = (2) − (1)	(4)	(5)	(6) = (5) − (4)	(7) = (6) − (3)	(8)	
Blood pressure checked monthly	55.6	74.2	18.6	26.7	40.0	13.3	−5.3	0.87	0.52
Weight checked monthly	95.7	90.5	−5.2	79.2	88.2	9.0	14.2	0.17	<0.01
Blood test taken	81.7	89.5	7.8	59.9	97.6	37.7	29.9	0.17	<0.01
Knowing blood type	15.6	18.7	3.0	10.7	23.7	13.0	10.0	0.55	0.05
Urine test taken	23.9	26.3	2.5	24.3	86.7	62.4	59.9	0.1	<0.01

Source: authors’ calculation. Note: Adj. DiD = adjusted DiD; column 8 $\hat{\beta_{3}}$ coefficients from regression model (1) predicting DiD of prenatal care behaviour after controlling for other covariates, including age, ethnicity, education, working status, household income, household size, number of children, and health status. The *p*-values are corrected for multiple comparisons using FDR correction.

**Table 5 healthcare-11-02407-t005:** Proportion of women taking supplements during pregnancy.

Supplement	Control			Intervention			Raw DiD	Adj. DiD	
	Before	After	Diff.	Before	After	Diff.		Diff.	*p*-Value
	(1)	(2)	(3) = (2) − (1)	(4)	(5)	(6) = (5) − (4)	(7) = (6) − (3)	(8)	
Folic acid	72.4	76.1	3.7	62.6	82.0	19.4	15.7	0.3	<0.01
Iron	96.3	99.1	2.9	96.5	99.5	3.1	0.2	0.14	0.09
Calcium	84.2	76.9	−7.3	60.2	91.5	31.3	38.6	0.11	<0.01
DHA	72.7	56.5	−16.2	56.8	62.5	5.7	21.9	0.3	<0.01

Source: authors’ calculations. Note: Adj. DiD = adjusted DiD; column 8 presents $\hat{\beta_{3}}$ coefficients from regression model (1) predicting DiD of supplement intake after controlling for other covariates, including age, ethnicity, education, working status, household income, household size, number of children, and health status. The *p*-values are corrected for multiple comparisons using FDR correction.

**Table 6 healthcare-11-02407-t006:** mMoM’s impacts on the health status of pregnant women.

Self-Reported Health Status	Control			Intervention			Raw DiD
	Before	After	Diff.	Before	After	Diff.	
	(1)	(2)	(3) = (2) − (1)	(4)	(5)	(6) = (5) − (4)	(7) = (6) − (3)
Very good	1.7	0.4	−1.3	1.0	1.6	0.6	1.8
Good	42.3	39.0	−3.3	56.2	72.1	15.9	19.2
Average	52.3	60.3	8.0	41.2	26.2	−15.0	−22.9
Poor	3.8	0.4	−3.4	1.6	0.1	−1.5	1.9

Source: authors’ calculations.

## Data Availability

Not applicable.

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
