# Peer review of "A Cluster Randomised Control Trial of an SMS-Based Intervention to Promote Antenatal Health amongst Pregnant Women in a Remote, Highland Region of Vietnam"

_healthcare, 2023, doi:10.3390/healthcare11172407_

Round 1
Reviewer 1 Report
Thank you for the opportunity to review the manuscript titled “ A cluster randomised control trial of an SMS based intervention to promote antenatal health amongst pregnant women in a remote, highland region of Vietnam”.
The authors conducted very interesting research that contains a lot of valuable data.
The following are my comments describing these issues.
1. At this point, the article is national in character and more appropriate for a national journal than an international one.
2. Material and Methods
Line 90-124 Please present the inclusion/exclusion criteria in the form of a diagram, it will be clearer.
3. Discussion
The discussion is not a discussion, but a repetition of the results.
Please move the table from the discussion to the results.
In the discussion, the authors should have compared their results with the results of other publications.
If authors look at PubMed once more and just type "pregnant AND sms" they will get publications from Kenya, Tanzania, Australia, China, and other countries...etc.
Line 300-320 Study findings in perspective
The authors do not describe in the discussion where they were looking for articles, because this is not a systematic review article.
From this part, the cited articles should be used and described in the discussion.
The discussion must be improved.
Author Response
Dear Reviewer,
We have addressed all comments and modified the manuscript. Please find details in the attached file.
We would like to thank you for your thoughtful comments and suggestions on the earlier draft of this paper.
Best regards
Author team

Reviewer 2 Report
The authors (AA) aim to assess the impact of the mMoM intervention on the MNCH (Maternal, Newborn and Child Health) knowledge, behaviour, and self-reported health status of pregnant women living in a remote, highland region of Vietnam. This is an article useful to increase our knowledge of the issue.
Addressing the following issues can make this manuscript eligible for publication.
Introduction
The references are appropriate, but AA should add other references about their topic regarding pregnant women in low-resource and underserved areas in other countries as reported below:
⁻ Benski AC, Schmidt NC, Viviano M, Stancanelli G, Soaroby A, Reich MR. Improving the Quality of Antenatal Care Using Mobile Health in Madagascar: Five-Year Cross-Sectional Study. JMIR Mhealth Uhealth. 2020 Jul 8;8(7):e18543. doi: 10.2196/18543.
⁻ Paduano S, Incerti F, Borsari L, Benski AC, Ernest A, Mwampagatwa I, Lilungulu A, Masoi T, Bargellini A, Stornelli F, Stancanelli G, Borella P, Rweyemamu MA. Use of a mHealth System to Improve Antenatal Care in Low and Lower-Middle Income Countries: Report on Patients and Healthcare Workers' Acceptability in Tanzania. Int J Environ Res Public Health. 2022 Nov 20;19(22):15342. doi: 10.3390/ijerph192215342.
Line 39: (Define acronym “mMoM”)
Methodology
Paragraph 2.2: Since AA have reported that they performed a cRCT AA should better explain the randomization.
Results
Lines 183-184: AA should report in the text the relevant data of the appendix tables.
Table 5: It is not clear the result about calcium after intervention.
Discussion
Lines 221-223: AA should soften this first inference in light of their findings.
The discussion should be more contextualised. AA should discuss their findings also by comparing these data with other studies including a similar population in other countries as well.
Minor editing of English language required
Author Response
Dear Reviewer,
We have addressed all of your comments and modified the manuscript. Please find details in the attached file.
Thank you very much for your thoughtful comments and suggestions on the earlier draft of this paper.
Best regards
Author team

Reviewer 3 Report
The study shows that mobile messaging-based interventions can significantly improve maternal health and care-seeking amongst women residing in a hard-to-reach areas in Vietnam.
First of all, some edition of English is required. Examples:
. "to overcome some these challenges"
. "in the mountainous in Northern Vietnam"
. "based behaviour change interventions can"
. "among pregnant woman"
The use and abuse of acronyms in the text makes it harder to follow and understand. As there is an appendix with the figures of the study, it would not be a bad idea to list all the acronyms used.
Although technical and statistical analysis seems coherent or plausible, formal aspects must be checked and completed. First of all, conclusions are missing. After the discussion, "study findings in perspective", "added value of this study", and "implications for policy, practice and research" have to be listed. Further details about the policy of privacy concerning personal data would also be appreciated.
The study shows that mobile messaging-based interventions can significantly improve maternal health and care-seeking amongst women residing in a hard-to-reach areas in Vietnam.
First of all, some edition of English is required. Examples:
. "to overcome some these challenges"
. "in the mountainous in Northern Vietnam"
. "based behaviour change interventions can"
. "among pregnant woman"
The use and abuse of acronyms in the text makes it harder to follow and understand. As there is an appendix with the figures of the study, it would not be a bad idea to list all the acronyms used.
Although technical and statistical analysis seems coherent or plausible, formal aspects must be checked and completed. First of all, conclusions are missing. After the discussion, "study findings in perspective", "added value of this study", and "implications for policy, practice and research" have to be listed. Further details about the policy of privacy concerning personal data would also be appreciated.
Author Response

(The authors gave the same response as above.)

Round 2
Reviewer 1 Report
The authors responded to all the reviewer's comments in a matter-of-fact manner.
The manuscript was revised according to the reviewer's comments.
Thank you.
Author Response
Dear Sir,
Thank you very much for your comments.
Best regards,
Author team
Reviewer 2 Report
The authors (AA) have addressed some reviewers' comments. Overall the changes made have improved the manuscript but there are a few points to improve.
Lines 245-246: it is not clear the type of association obtained by AA. Clarify it.
Paragraph 4.1 Study findings in perspective: AA should not report in the discussion section (par. 4.1) where they were searching for articles, because this is not a systematic review. The cited papers should be used and described in the discussion.
Supplementary tables: since the tables should be understandable without viewing the test, AA should explain the meaning of “income” and “Dpost” in the notes.
Minor editing of English language required
Author Response
Dear Sir,
Thank you very much for your comments. Please find our revision in the attached files.
Best regards,
Author team
